# Associations between Home- and School-Based Violent Experiences and the Development of Sexual Behavior in Young Adolescent Girls in the Rural Southern Region of Malawi

**DOI:** 10.3390/ijerph19105809

**Published:** 2022-05-10

**Authors:** Sadandaula Rose Muheriwa Matemba, Rosina Cianelli, Natalie M. Leblanc, Chen Zhang, Joseph De Santis, Natalia Villegas Rodriguez, James M. McMahon

**Affiliations:** 1School of Nursing, University of Rochester, 601 Elmwood Avenue, Rochester, NY 14642, USA; natalie_leblanc@urmc.rochester.edu (N.M.L.); chen_zhang@urmc.rochester.edu (C.Z.); james_mcmahon@urmc.rochester.edu (J.M.M.); 2School of Nursing and Health Studies, University of Miami, Coral Gables, FL 33146, USA; rcianelli@miami.edu (R.C.); jdesantis@miami.edu (J.D.S.); 3School of Nursing, The University of North Carolina at Chapel Hill, Chapel Hill, NC 27599, USA; navilleg@email.unc.edu

**Keywords:** adolescents, home-based violence, school-based violence, sexual behavior

## Abstract

Studies show that adolescent girls who experience violence grow up with fear and develop survival mechanisms that increase their susceptibility to sexually transmitted infections including HIV. However, the relationship between violence and the development of sexual behavior in young adolescent girls is under-investigated. We examined the Malawi Schooling and Adolescent Study data to explore the associations between home- and school-based violence and sexual behaviors in 416 young adolescent girls in rural Southern Malawi. Bivariate Logistic Regression analysis was applied to determine associations. Of 353 (84.9%) girls who had sex with a male partner, 123 (34.8%) experienced home-based violence, and 53 (15%) experienced school-based violence. The odds of girls who experienced home-based violence (OR = 2.46, 95% CI = 1.21, 5.01) and those who first experienced home-based violence between 13 and 14 years (OR = 2.78, 95% CI = 1.35, 5.74) were higher among girls who had multiple sexual partners than those with a single sexual partner. With school-based violence, sexual initiation, having multiple sexual partners, and not using protection were positively associated with experiencing teasing, sexual comments, punching, and touching in private areas in transit to school and by a teacher. These results suggest that home- and school-based violence should be essential components of research and biobehavioral interventions targeting the sexual behaviors of young adolescent girls.

## 1. Introduction

Every year, an estimated 1 billion children and adolescents experience violence globally [1], and 10% of all deaths that occur among adolescent girls are due to violence [2,3]. The United Nations Children’s Fund report that violence accounts for a more substantial proportion of deaths as children move from early childhood to adolescence. Among girls, the violence-related deaths (out of all causes) rise from 0.4% at age 0–9 to 4% at age 10–14 and to 13% at age 15–19 [2]. In 2015 alone, 82,000 adolescents died due to violence [4]. Studies show that the nature, quantity, and impact of violence victimization vary across childhood and adolescence with the different capabilities, activities, and environments characteristic of different stages of development [5,6].

While all adolescents experience violence, being a girl presents a unique vulnerability with life-long consequences [7,8]. Gender discrimination in tandem with cultural norms and practices means that adolescent girls are likely to experience certain forms of violence, such as sexual violence, at much higher rates than boys [9]. Girls are also more likely to be exposed to certain harmful practices, such as genital mutilation and early and forced marriage which obligates them to participate in harmful and non-consensual sexual behaviors that increase their vulnerability to adverse sexual and reproductive health outcomes [10]. Puberty intensifies the vulnerability of girls to violence because, as the girls transition into adulthood, sexuality and sex-based gender roles begin to assume greater importance in how adolescent girls are viewed socially [9,10]. Consequently, for girls (7–55%) globally, the first experience of sexual intercourse is coerced [1,9].

For most adolescents, the home environment provides a positive, nurturing, affirming, and loving context [2]. However, home is also where young adolescents’ first exposure to violence mostly occurs [4]. Data show that the most common form of violence that adolescents experience is physical violence within the context of discipline, usually in their own homes and at the hands of their caregivers [11,12]. Among younger adolescent girls aged 10 to 14, nearly two out of three are subjected to physical punishment regularly, with country rates ranging between 45 and 95 percent [11,12]. Home-based violence, which mainly takes the form of hitting, slapping, whipping, or verbal abuse, is commonly exerted by caregivers while teaching young adolescents self-control and acceptable behavior [2,12]. However, these forms of violence are considered harmful and have been shown to increase the likelihood that young adolescents adopt unhealthy coping strategies to alleviate their distress [6,13]. Sexual violence has also been reported in homes, perpetrated by caregivers and others affiliated with the home setting [14]. In Malawi, a recent survey showed that 18.8% of girls 13–17 years old experience sexual violence in the home, and 12.4% report the perpetrator to be a family member [15].

The violence young adolescents experience in their homes is reported to have an impact on their sexual behaviors. Mmari, Kalamar, Brahmbhatt, and Venables [16] found that in Baltimore and Johannesburg, female adolescents exposed to home-based violence were more than twice as likely to have had sex compared to those who had not. Garza-Torteya in a study among high school adolescents 15–19 years old found that parental affection was associated with a delayed sexual debut and a lower number of sexual partners [17]. However, when the parents used physical punishment, the adolescents had an earlier sexual debut, multiple sexual partners, little or no use of contraception, and an increase in having sex under the influence of drugs or alcohol [17]. In contrast, a longitudinal study in Kenya showed that adolescents (12–19 years old) who were spanked or slapped by parents in the home as a form of discipline were less likely to report sexual debut [18].

Violence among girls also occurs in places where young people are meant to learn and socialize [4]. Studies have revealed that violence in school-based settings is a daily reality that denies millions of young people the fundamental human right to education. Estimates show that 246 million children and adolescents experience violence in and around school globally every year [19]. The types of violence in schools include peer bullying, sexual harassment, mental and physical abuse often exerted as a means of corporal punishment [19]. Adolescents’ experience of sexual assault perpetrated by peers and teachers is also reported to occur at an alarming frequency on school premises [20]. In the 28 countries with available data, 9 in 10 adolescent girls who have reported forced sex said it occurred for the first time at the hands of someone close or known to them, and for school-based violence, classmates, teachers, and friends were among the most frequently cited perpetrators of the latest incident in five countries with comparable data, that is Cambodia, Haiti, Kenya, Malawi and Nigeria [2]. Another nationally representative survey in Malawi revealed that classmates and schoolmates were the most reported perpetrators of sexual violence in 31.8% of the girls 13–17 years old [15]. Nevertheless, the impact of violence on sexual initiation has been shown to differ based on the developmental age at which it occurs. Warner and Warner [6], in secondary data analysis of 10,070 youth from four waves of the National Longitudinal Study of Adolescent to Adult Health (Add Health), found that youth victimized in late adolescence displayed an accelerated trajectory of sexual activity. While youth victimized in early adolescence were less likely to debut or engage in sexual risk behaviors but were more likely to engage in other deviant behaviors such as violent perpetration.

In Malawi, recent national surveys show that 72% of female adolescents and children 14 years and younger experienced physical violence in the home, including hitting on the head, ears, or face or hitting the child hard and repeatedly [21]. It is also estimated in Malawi that 70% of youth experience violence in primary schools [22]. However, few studies have examined the association between violence and the development of sexual behavior. Given the high prevalence of violence against young girls in Malawi, this knowledge is necessary to increase public health awareness and develop interventions to prevent violence among young adolescent girls and reduce the adverse outcomes of violence. Besides, adolescent girls are at the nexus of violence against children and violence against women and understanding the complexity of their lives in the home and in and around the school is crucial for improving interventions to respond to and prevent both forms of violence [10]. This study examined the associations between home- and school-based violence and the development of sexual behavior in adolescent girls 14 years old in the rural southern region of Malawi.

## 2. Methods

### 2.1. Study Design and Setting

This study analyzed the baseline data of 416 adolescent girls enrolled in the Malawi Schooling and Adolescent Study (MSAS) in the rural Southern region of Malawi [23]. The girls were 14 years old and were enrolled in grades 4 to 8. The Malawi Schooling Adolescent Study was a six-round longitudinal study of 2649 boys and girls ages 14–17 years who were both enrolled and not enrolled in school in two rural districts, Balaka and Machinga of the Southern Region of Malawi. We limited the current study to 14-year-old girls because this was the youngest age group in the parent study. We focused on early adolescents, an understudied group because sexual development and behavior change rapidly between the ages of 14 to 17 and findings might vary across age groups. The study was conducted by the Population Council of New York and the University of Malawi from the year 2007 to 2013. The study explored the mechanisms linking school quality and educational outcomes to sexual behavior and the acquisition of HIV and herpes simplex-2 (HSV-2). Participants were randomly selected from the enrollment rosters at 59 selected primary schools in the two districts. A total of 885 boys and girls who were not enrolled in school were drawn from the communities surrounding the selected schools. Data were collected in structured face-to-face interviews, and sensitive information including sexual behavior and violence were obtained through the computerized audio interview method.

### 2.2. Measures

Sexual behavior was the primary outcome and had three indicators: sexual initiation, engagement of multiple sexual partners, and the use of protection from pregnancy or STIs. Sexual initiation was ascertained by asking “What age were you the first time you had sex?” The response was continuous; 0 for those who have never initiated sex and the actual age of sexual initiation for those who had initiated sex. However, this variable was transformed and collapsed to a dichotomous variable and recoded: 0 for those who had never had sex and 1 for those who reported having had sex. Having had multiple sexual partners was ascertained by the question “In total, how many different people have you had sex with in your life?” The response was continuous, the girls who reported having had sex indicated the number of men they had ever had sex with. This variable was also collapsed into a dichotomous variable and was coded 1 for those who had one sexual partner and 2 for those who reported having had multiple sexual partners, ranging from 2 to 5. The use of protection was ascertained by the question “The very first time that you had sex, did you or the other person use any method to prevent pregnancy or a sexually transmitted disease?” The response was dichotomous, no = 0 and yes = 1.

Our main independent variables were home- and school-based violence. Home-based violence was assessed by asking the adolescents two questions. The first question was “Has anyone in your household ever punched, slapped, or whipped you?” The response option was dichotomous, no = 0 and yes = 1. The adolescents who answered yes were asked to indicate the age they were first punched, slapped, or whipped in the home, and the adolescents were to indicate the age they first experienced this type of violence in the home. This variable was also collapsed into a dichotomous variable and coded 1 for those who had first experienced violence at an earlier age between 6 and 12 years, and 2 for those who had first experienced home-based violence at an older age between 13 and 14 years. School violence was ascertained by asking the adolescents how often they had experienced the following types of violence on the way to school or in school perpetrated by the schoolmates or the teachers in the current school year: teased or upset (emotional or psychological violence), sexual comments made to them (sexual harassment), punched, slapped, or whipped (physical violence), or touched or pinched on the breasts, buttocks, or genitalia (sexual violence). The responses were categorized as never = 0, a few times (once to five times) = 1 and many times (6 times or more) = 2. We added tribal affiliation (Yao, Lhomwe, Chewa, and others), religion, participation in religious activities, and HIV risk perception to the analysis to further explore the associations of sexual behavior in young adolescents because evidence shows that these factors are associated with the development of some forms of sexual behavior. These variables were all coded as dichotomous variables.

### 2.3. Analysis

We analyzed the missing values and determined that overall, 3.1% of the values were missing. Given that <5% missingness is considered a lower threshold below which missing data handling techniques, such as multiple imputation, provide little benefit [24,25], we applied listwise deletion, and the excluded cases ranged from 25–45. In addition, several variables related to sexual behavior had a substantial number of non-responses due to skip patterns for non-applicable questions (e.g., only girls who reported ever having sex were asked about protection at first sex). We conducted descriptive statistics and used frequencies and percentages to characterize our variables. We used bivariate logistic regression to explore the association between home- and school-based violence and sexual behavior. The relationship between the predictor and outcome variables was estimated using odds ratio with a 95% confidence interval. We did not employ the adjustments to control for experiment-wise Type 1 error due to the exploratory nature of the analysis. We report exact *p*-values, and we avoided using *p*-value cut-offs and wording related to the concept of statistical significance following the recent guidelines advocated by the American Statistical Association [26]. Instead, we used the Odds Ratio (OR) associated with an increase in exposure to sexual behavior [27] and interpretation of the confidence interval [28]. We used IBM SPSS (version 26) to conduct all the analyses.

## 3. Results

Table 1 presents the descriptive analyses of the demographic characteristic and home- and school-based violence experiences of young adolescents. Most participants were Yao and predominantly Christians. About 136 (35.6%) of adolescents reported experiencing home-based violence, and on average 57 (17.5%) reported experiencing school-based violence. Most of the young adolescent girls reported experiencing school-based violence a few times (once or twice), and 3.2% of the young adolescent girls experienced school-based violence many times and almost every day. The young adolescent girls were teased more in school by their schoolmates, while punching, slapping, or whipping was perpetrated more by the teachers in the school. More sexual comments and touching or pinching of the breasts, buttocks, or genitalia were perpetrated by schoolmates.

Table 2 shows the sexual behaviors of young adolescents. The results show that 353 (84.9%) of the girls reported having a sexual encounter with at least one sexual partner. However, less than one-quarter of these girls reported the age they first had sex and the events surrounding their first sexual encounter. The mean age of sexual initiation was 11.9 ± 2.7 years. Out of 353 girls who reported having had sex with at least one male partner, 123 (34.8%) experienced home-based physical violence, and 53 (15.1%) experienced school-based violence.

Table 3 presents the results of the associations between violence and sexual behavior. When subscales were entered into the model as independent predictors home-based violence was positively associated with engaging multiple sexual partners. We found a positive association between home-based violence and having multiple sexual partners. The odds of engaging multiple sexual partners among the girls who reported being punched, slapped, or whipped in the home (OR = 2.46, 95% CI = 1.21, 5.01) were 2.46 times higher than among the girls who were never punched, slapped, or whipped in the home. Also, the odds of engaging multiple sexual partners among the girls who were punched, slapped, or whipped in the home for the first time when they were within the ages of 13 and 14 (OR = 2.78, 95% CI = 1.35, 5.73) were more than two times higher than among the girls who were punched, slapped or whipped in the home for the first time when they were between the ages of 6 to 12 years. However, based on the wide confidence intervals, our findings were inconclusive regarding the associations between home-based violence and sexual initiation and the use of protection, such as condom and contraceptive use.

On the other hand, some forms of school-based violence were associated with sexual initiation, having multiple sexual partners, and using protection against pregnancy or STIs during first sex. The results showed that the odds of sexual initiation among girls who were teased or upset in transit to school (OR = 1.59, 95% CI = 1.14, 2.21), who reported being teased by a teacher (OR = 2.09, 95% CI = 1.05, 4.15), and who reported being touched in the private areas by a teacher (OR = 4.35, 95% CI = 1.92, 9.88) were higher compared with girls who never experienced these forms of violence. The odds of engaging multiple sexual partners among the girls who experienced sexual comments from a teacher were double (OR = 2.00, 95% CI = 1.05, 3.80) those of girls not reporting such comments; whereas the odds of using a condom or hormonal contraceptive at first sex were 39% lower (OR = 0.61, 95% CI = 0.11, 0.89) among girls who were teased or upset in transit to school compared to those who were not.

In contrast, we found the effect of some forms of violence counter intuitive. The odds of sexual initiation of the adolescent girls who were punched, slapped or whipped by a teacher (OR = 0.47, 95% CI = 0.23, 0.93) and those who experienced sexual comments on the way to school (OR = 0.50, 95% CI = 0.27, 0.91) were 53% and 50% lower, respectively, compared to girls not reporting these sexual violence experiences.

Among the sociodemographic characteristics, we found that religion and HIV risk perception were associated with sexual behavior. The odds of engaging multiple sexual partners of adolescent girls who reported participating in a religious youth group were 61% lower (OR = 0.39, 95% CI = 0.17, 0.91) compared to girls who were not in a religious youth group. The odds of using protection were nearly four times higher (OR = 3.90, 95% CI = 1.16, 13.11) among girls who perceived a greater chance of contracting HIV in the future compared to girls reporting less perceived HIV risk. Nevertheless, based on the wide confidence intervals, our findings were also inconclusive regarding the associations between religious affiliation and participating in a choir and the use of a condom or hormonal contraceptives to prevent pregnancy and STIs.

## 4. Discussion

Violence is an experience that can negatively affect every aspect of young people’s lives, including their sexual health. Our findings on the association between home- and school-based violence and the development of sexual behavior vary and depend on the type of violence, the age the violence occurred to the adolescent, and the perpetrator of the violence. This finding is consistent with what Finkelhor and colleagues [5] pointed out, that nature, quantity, and impact of violence victimization vary across childhood and adolescence with the different capabilities, activities, and environments characteristic of different stages of development. Studies have also shown that vulnerability to violence varies across the age spectrum. The rate of victimization increases through the teenage years, peaks at around age 20, and steadily decreases through the remaining years [5,29,30], suggesting the need for increased efforts for screening for violence victimization among young adolescents.

This study revealed evidence of the association between experiencing home-based violence, particularly when the girls started experiencing the home-based violence when they were between 13 and 14 years and had multiple sexual partners. However, we did not find enough evidence to support the association between home-based violence and sexual initiation. The finding that physical violence was associated with having multiple sexual partners when the girls were older is consistent with Warner and Warner [6], who found that youth victimized in late adolescence displayed an accelerated trajectory of sexual activity as compared to youth victimized in early adolescence, who were less likely to debut or engage in other sexual risk behaviors. This finding relates to the role of perception on the influence of sexual behavior, which varies with age [6,31]. Research shows that there is an age-differentiated response to violence where older adolescent victims of violence display an overinvestment response, associated with rapid entry into dating relationships and faster progression from dating into co-residential unions [6]. Adolescent victims of violence who experience overinvestment or anxious attachment-related responses to their victimization are often propelled into precocious intimate relationships and are at a greater risk of engaging in precocious sexual activity [32,33]. Thus, anxiously attached adolescents are more likely to engage in sexual activity, perhaps due to a desire for intimacy and closeness [6]. Therefore, frequent screening for violence and preventing violence in the home can help prevent risky sexual behaviors in adolescents.

The finding that home-based violence was not associated with multiple sexual partners when initiated between the ages of 6 and 12 years can be attributed to what Warner and Warner [6] identified as a withdrawal response to violence common among young adolescents. Studies show that young adolescents elicit anger in response to violence victimization [34], and adopt a hostile attribution bias toward intimate relationships, which leads to shunning intimacy altogether [32,35]. This finding shows that violence among young adolescents might have a different meaning and consequences from that of older adolescents, though still detrimental and require more studies to inform age-appropriate strategies and interventions to end violence among young adolescents.

The fact that home-based physical violence was not conclusively associated with sexual initiation might be associated with inconsistencies in reporting sexual behavior among young adolescents in this study. For example, when asked at what age the girls first had sex, less than a quarter of the girls indicated the age they had sex and three-quarters indicated that they had never had sex. Whereas, when asked the total number of men they had had sex with in their life, 84.9% indicated having had a sexual encounter. This inconsistency might affect the quality of evidence of the results. On the other hand, the inconclusive results of the association between home-based violence and sexual initiation might suggest that the violence was a corporal punishment meant to correct the adolescent girls’ behavior, which, though illegal according to the constitution of Malawi [36], is universally accepted as a usual means of disciplining children [37]. As such, one might argue that violent discipline might not have affected adolescents in a manner where they would seek emotional comfort in a romantic relationship, as the literature suggests [32]. However, this study did not investigate why the adolescents were punched, slapped, or whipped in the home. Therefore, larger and more robust studies are needed to determine the causes of violence and how physical violence affects young adolescent girls.

Though inconclusive, this finding also supports the long-standing need for interpretation, clarity, and prohibition for the near-universal acceptance of corporal punishment in childrearing in the law in Malawi [21,37,38,39]. Much as the study did not show enough evidence on the association between physical violence in the home and early sexual debut among adolescents, previous studies have shown that violence victimization in early adolescents is associated with other deviant behaviors such as criminal behaviors, drug and substance use, and depression in later lives which can influence risky sexual behaviors [6,16]. Therefore, more research is needed to inform legal frameworks that can support guidelines and measures to enforce laws to end any physical violence in the home including corporal punishment. Having other professionals designated as mandatory reporters of violence against children and young adolescents in Malawi can also help in the fight against violence among young people.

In this study, we also found that being teased on the way to school, being teased by the teachers, and being touched on the breasts, buttocks, and genitalia by the teachers were positively associated with initiating sex. While being punched by the teacher and sexual comments experienced on the way to school were negatively associated with sexual initiation. Concerning the engagement of multiple sexual partners, the study found that only sexual comments by the teachers were associated with engaging multiple sexual partners. Meanwhile, with the use of protection, the results showed that being teased on the way to school was associated with lower odds of using protection and being punched by schoolmates was associated with higher odds of using protection (though inconclusive). These results suggest that the safety of young adolescents when going to or coming from school and teacher–student interaction in schools are crucial determinants of adverse sexual behavior in young adolescents in Malawi. This finding is aligned with studies in Botswana, Malawi, and Mozambique that showed that schools are insecure for young girls [40,41,42]. A recent study in Malawi also revealed that girls are vulnerable to HIV infection in part because the social systems in which they live have failed to protect them [42], suggesting the need for increased efforts to end school-based violence.

While the authors acknowledge the Malawi government’s efforts in making schools safer for young adolescent girls, enforcement still proves difficult and slow. Speedy measures, policies, and guidelines should be put in place to ensure the young adolescent girls’ safety when going and coming from school. In Malawi, parents do not escort their children to school, particularly in the rural areas, and there is no proper transportation system for children and young adolescents going to school. Therefore, the finding that the young girls’ experiences of physical and sexual harassment in transit between home and school were positively associated with sexual initiation and other risky sexual behaviors are essential for informing a multidisciplinary approach to initiating programs and putting in place appropriate transport systems that will ensure safety for girls and all children. Thus, interventions to reduce violence among adolescents should target their safety on the roads to and from school and their interaction with teachers as well as with schoolmates when they are in school. There is also a great need for community sensitization to protect children as they go to or come from school. Additionally, young adolescents should be sensitized to the signs of impending violence and be empowered to avoid and report any suspicious behaviors. School violence is not uncommon in Malawi, but its impact on young adolescent girls has not been identified as a health priority. Strengthening violence and sexual behavior development courses in school curriculums may increase awareness of violence in tandem with emerging sexuality as a developmental issue among adolescents, and ultimately inform the practice and prevention efforts.

The finding for other forms of violence were counter-intuitive; that they were protective of risky sexual behavior might suggests that not all situations of violence were equally perceived as sexually provoking by girls, implying that a combination of factors in the young adolescent girls’ environment contribute to their sexual behavior development. This finding aligns with what Gådin [43] described about sexual harassment, that it is a contradictory, complex, and multilayered behavior leading to a reaction depending on who the harasser is, frequency, and where it occurs. Thus, not every exposure to physical and sexual harassment and violence has negative consequences for individual girls. Nevertheless, being in an environment where there is a risk of being exposed to harassment leads to a hostile environment and suggests the need to develop methods to make homes and schools safe places for adolescent girls. However, these data were limited in the degree to which they can explain why some forms of violence had no association with sexual behavior. Future research may help to understand these types of relationships.

Consistent with the notion that a combination of factors plays a role in developing sexual behavior in young adolescents, this study found that religion and HIV risk perception were associated with multiple sexual partners and the use of protection. The study found that participating in a religious youth group was negatively associated with the engagement of multiple sexual partners, while being a Muslim and participating in a church choir, though the results were inconclusive, were negatively associated with using a condom or hormonal contraceptive during the first sex. These findings might be explained by the role of religious norms for adolescent girls in Malawi, which many adolescents internalize before becoming sexually active. Consequently, their religious norms and beliefs shape their sexual behavior [44].

Consistently, the Islamic faith prohibits premarital sex, which is equated to committing Kibirah, one of Islam’s greatest sins [45]. Thus, any sex outside marriage is a criminal offense and subsequently prohibited in Islam [45]. Therefore, the girls might not use protection because they were not engaging in sexual activities. Moreover, previous studies show that religion is a protective factor against multiple sexual partners [46,47]. These findings demonstrate the opportunity religious institutions have to reach young adolescents with sexual health information and entail the need for collaboration with religious institutions in interventions for young adolescent girls’ sexual health. Nevertheless, this study gathered little evidence to explain why participating in youth groups was negatively associated with having multiple sexual partners, and why being in a choir and being a Muslim were associated with not using protection the first time the girls had sex, suggesting the need for further qualitative research to understand the role of religion in the development of sexual behaviors in young adolescent girls.

The finding that the odds of the young adolescents who perceived a greater risk of contracting HIV in the future were nearly three times higher among the girls who used protection at the first sexual encounter may not be surprising. Studies have shown that an increase in HIV risk perception is associated with increased condom use and use of pre-exposure prophylaxis compared to unchanged risk perception [48,49]. Although several factors determine the use of protection among adolescents, these results suggest that increasing cognitive interventions among young adolescents can improve their sexual behaviors.

### Limitations

This study only analyzed the data of the first round of the Malawi Schooling and Adolescent longitudinal study. The cross-sectional study design precludes making solid causal inferences regarding the observed associations. Thus, the analysis should be considered exploratory, and we therefore did not apply a multiple test adjustment procedure. In addition, all data are based on self-report, and are thus subject to various forms of response bias, including social desirability and inaccurate recall. Additionally, the study only investigated adolescent girls who were enrolled in school. Thus, the most vulnerable adolescents may have been left out of this analysis. Further, the study only examined home-based physical violence and did not investigate other forms of home-based violence such as sexual violence and emotional trauma. Additional studies are needed to explore other forms of home-based violence, determine strategies to ensure the safety of young adolescent girls within the family institutions, and determine the young adolescents’ knowledge of impending violence and how to prevent them. Such studies can help inform strategies and interventions that can empower young adolescents to identify and report experiences of violence. The study also focused on those who were 14 years old and used older data. However, though older data were used, the results revealed important determinants of risky sexual behavior among young adolescents that will inform future research on adolescents younger than 14 years to better inform the strategies to end violence in young adolescents in Malawi.

## 5. Conclusions

This study has shown that young adolescent girls in rural Malawi are at an increased risk for adverse sexual behavior due to the violence they experience in homes and within the school premises. Although a combination of factors in the young adolescent girls’ environment determines their sexual behaviors, the findings of this study show that violence among young adolescent girls has a significant contribution to the development of sexual behaviors of young adolescent girls. Therefore, there is a need for a multidisciplinary approach in the fight against violence among girls. By creating awareness and educating the public, it is possible to promote community and social responsibility, stop violence and help abused children and adolescents achieve their optimum potential.

## Figures and Tables

**Table 1 ijerph-19-05809-t001:** Sociodemographic and violence characteristics of the young adolescents (*N* = 416).

Variable Characteristics	*n*	*%*	*n*	*%*	*n*	*%*
Sociodemographic Characteristics						
Tribe						
Yao	156	37.5				
Chewa	83	20.0				
Lhomwe	111	26.7				
Other	73	17.5				
Religion						
Christians	251	60.3				
Moslems	165	39.7				
Participation in religious activities						
Choir	153	36.9				
Youth group	134	32.3				
Prayer meeting	125	30.0				
Night prayers	61	14.7				
HIV Risk Perception						
Worry of contracting HIV infection currently (*n* = 392)						
Perceived no worry	267	68.1				
Perceived worry	125	31.9				
Perceived chances of contracting HIV in the future(*n* = 402)						
Perceived no chance	183	45.5				
Perceived a chance	289	54.5				
Home-based violence						
Ever punched, slapped, or whipped in the home						
No	246	64.4				
Yes	136	35.6				
Age first punched, slapped, or whipped in the home						
6–12 years	70	65.4				
13–14 years	37	34.6				
School-based violence	On the way to school		By Schoolmates		By Teachers	
Teased or upset						
Never	303	72.8	285	65.5	345	82.9
Few times	15	3.6	115	27.6	61	14.7
Many times	98	23.6	16	3.8	10	2.4
Punched, slapped, or whipped						
Never	241	76.3	327	78.6	305	73.3
Few times	64	20.2	77	18.5	100	24.0
Many times	11	3.5	12	2.5	11	2.6
Sexual comments						
Never	313	75.2	322	77.4	357	85.8
Few times	72	17.3	83	20	49	11.8
Many times	31	7.5	11	2.6	10	2.4
Touched on breasts, buttocks, or genitalia						
Never	359	86.3	355	85.3	377	90.6
Few times	48	11.5	49	11.8	29	7.0
Many times	9	2.2	12	2.9	10	2.4

**Table 2 ijerph-19-05809-t002:** Sociodemographic characteristics, violence and sexual behaviors of the young adolescents (*N* = 416).

Sexual Behavior	*n*	*%*
Ever had sex		
Never	63	15.1
Had one or more sexual encounter	353	84.9
The total number of people ever had sex with (*n* = 353)		
One sexual partner	288	81.6
Multiple sexual partners (2–5 different men)	65	18.4
Age of sexual initiation (*n* = 81)		
6–10 years	21	25.9
11–14 years	60	74.1
Type of relationship with the person had first sex with (*n* = 70)		
Spouse	8	11.4
Fiancé (a promise to marry)	6	8.6
Boyfriend	32	45.7
Acquaintance	6	8.6
Hit and run	3	4.3
Relative	8	11.4
Someone else	7	10.0
Initiated sex as part of traditional initiation (*n* = 353)		
No	334	94.6
Yes	19	5.4
Used protection at first sex		
Not used protection	321	90.9
Used protection	32	9.1
Use of protection to prevent pregnancy, HIV, and other STIs (*n* = 32)		
Pill	2	6.3
Injectable	4	12.5
Condom	24	75.0
Other	2	6.3

**Table 3 ijerph-19-05809-t003:** Bivariate associations between violence and sociodemographic characteristics and sexual behavior, (*N* = 416).

Violence Characteristics	Sexual Initiation					Multiple Sexual Partners					Use of Protection				
	*n*	OR	*p*	95% CI		*n*	OR	*p*	95% CI		*n*	OR	*p*	95% CI	
				LL	UL				LL	UL				LL	UL
Home-based violence															
Ever been punched, slapped, or whipped in the home	382	1.07	0.842	0.55	2.08	348	2.46	0.013	1.21	5.01	56	1.28	0.574	0.55	2.99
Age first punched, slapped, or whipped in the home	389	1.41	0.459	0.57	3.51	353	2.78	0.006	1.35	5.73	57	1.32	0.617	0.45	3.89
School-based violence															
Teased or upset on the way to school	382	1.59	0.006	1.14	2.21	349	1.04	0.838	0.70	1.56	57	0.61	0.011	0.11	0.89
Teased or upset by schoolmates	381	0.80	0.462	0.45	1.44	347	1.20	0.579	0.64	2.25	57	0.87	0.663	0.51	0.80
Teased or upset by a teacher	381	2.09	0.036	1.05	4.15	349	0.44	0.096	0.17	1.16	57	0.43	0.433	0.56	1.31
Punched, slapped, or whipped on the way to school	382	0.73	0.404	0.35	1.52	348	1.95	0.082	0.92	4.15	57	0.69	0.391	0.47	1.20
Punched, slapped, or whipped by the schoolmates	381	0.90	0.785	0.44	1.86	347	0.46	0.069	0.20	1.06	57	2.66	0.047	0.11	1.95
Punched, slapped, or whipped by a teacher	382	0.47	0.031	0.23	0.93	347	1.08	0.837	0.54	2.15	57	1.53	0.291	0.36	1.21
Sexual comments on the way to school	381	0.50	0.024	0.27	0.91	347	0.81	0.555	0.41	1.62	55	1.91	0.102	0.13	1.42
Sexual comments by the schoolmates	381	1.65	0.148	0.84	3.25	347	2.03	0.088	0.90	4.56	56	0.80	0.592	0.59	1.03
Sexual comments by a teacher	382	1.68	0.156	0.82	3.45	348	2.00	0.035	1.05	3.80	56	0.47	0.070	0.06	1.56
Touched or pinched on the breasts, buttocks, or genitalia on the way to school	381	1.32	0.568	0.51	3.39	347	1.08	0.877	0.42	2.78	56	0.47	0.166	0.31	1.82
Touched or pinched on the breasts, buttocks, or genitalia by the schoolmates	382	0.78	0.599	0.32	1.95	348	1.80	0.159	0.80	4.05	56	1.67	0.389	0.66	1.69
Touched or pinched on the breasts, buttocks, or genitalia by a teacher	380	4.35	<0.001	1.92	9.88	345	0.86	0.759	0.33	2.25	56	0.47	0.114	0.18	1.69
Sociodemographic characteristics															
Religion	389	0.94	0.939	0.48	1.83	353	0.79	0.703	0.23	2.70	56	0.42	0.047	0.01	1.74
Participation in religious activities															
Choir	389	0.88	0.702	0.46	1.69	353	2.03	0.126	0.82	5.02	57	0.43	0.040	0.04	1.67
Youth group	389	1.56	0.147	0.86	2.86	353	0.39	0.029	0.17	0.91	57	0.50	0.056	0.02	1.42
Prayer meetings	389	1.03	0.935	0.55	1.93	353	1.60	0.276	0.69	3.75	57	1.93	0.112	0.15	1.47
Night prayers	389	0.82	0.632	0.37	1.85	353	1.32	0.600	0.47	3.68	57	0.60	0.278	0.41	1.43
HIV risk perception															
Worry that might contract HIV	367	1.60	0.101	0.91	2.82	331	1.28	0.439	0.69	2.37	54	2.28	0.245	0.57	9.09
Chances of contracting HIV in future	376	0.75	0.292	0.43	1.29	340	1.18	0.597	0.64	2.16	57	3.90	0.027	1.16	13.11

## Data Availability

The data presented in this study are available on request from The Population Council New York.

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
