# Peer review of "Associations between Home- and School-Based Violent Experiences and the Development of Sexual Behavior in Young Adolescent Girls in the Rural Southern Region of Malawi"

_ijerph, 2022, doi:10.3390/ijerph19105809_

Round 1

Reviewer 1 Report

This is a relevant and interesting and generally well-developed paper. Some points can be improved, and I raise some suggestions:

  1. In the Methods section:
  • Please explain why the study only includes girls 14 years old (and not the whole sample of girls aged 14-17)
  1. In the Analysis section:

In page 4, rows 169-172, you state: “We analyzed the missing values and determined that, overall, 3.1% of the values 169 were missing completely at random. Given that <5% missingness is considered a lower 170 threshold below which missing data handling techniques, such as multiple imputations, 171 provide little benefit [24, 25], we applied listwise deletion, and the excluded cases ranged 172 from 25-45.”

However, it is not clear which variables are you referring to at this point. In fact, there are several variables with a very high proportion of missing values (Age of sexual initiation, Type of relationship with the person had first sex with and Use of protection to prevent pregnancy, HIV, and other STIs). And later, you took the right decision of not including them in the bivariate regressions… But you don´t make any comments on such decision. Please make it evident.

  1. Table 2:

- Given this table includes some variables with a high number of missing cases, those missing cases should be included in the table, and the percentages recalculated including them, to have a more accurate understanding of the data.

- What is the basis for describing a sexual onset between 6 and 10 years of age as "early" and when it occurs between 11 and 14 years of age as "older"? All of them seem quite early... Please comment and argue on this.

  1. Pages 6 & 7, in the comments referred to table 3, you do not specify which dependent variable are you referring to each time you comment on the value of an odd ratio… so it is confusing, and the comments are not well specified. Please review and rewrite this section making clearer the results.

  1. Limitations:

In the limitations section you state that you used older data, and this makes it evident that nowhere do you mention the date on which this survey was conducted, information that should be included in the methods section.

Author Response

Dear Reviewer,

Thank you very much for your comments. We appreciate your feedback. Please see the attachment.

Sincerely,

Sadandaula Rose Muheriwa Matemba

Reviewer 2 Report

I enjoyed the opportunity to review this manuscript on an important topic that warrants additional research and scholarship. This study examined the associations between home and school- based violence and the development of sexual behavior in adolescent girls 14 years old in the rural southern region of Malawi.

The paper is well-written and the authors have taken care in its presentation. The purpose is clearly stated, but not linked a theoretical significance of work (e.g., theory of reasoned action, social development model, ecological systems theory). The literature review is comprehensive with the use of relevant and up to date literature. The methods section is transparent and explicit,  providing sufficient details to allow the work to be reproduced. Interpretations are appropriate and justified in relation to analyses. Additionally, the authors acknowledged several limitations associated with the study and the design it employs. It is my recommendation to accept this manuscript in present form.  

Author Response

(The authors gave the same response as above.)

Reviewer 3 Report

Not clear what "out-of-school" means, line 127.  Does it mean they were not in school or the opposite?

"results inconclusive reported in lines 213, 260, & 390.  It is not clear why that is the case. 

In one instance, line 213, one group was 41% higher than the other.  Is that inconclusive?  Why? 

Is the term "results inconclusive" due to some statistical result or author interpretation, or what?

Author Response

(The authors gave the same response as above.)
